# OpenReview forum: "Reinforced Learning Explicit Circuit Representations for Quantum State Characterization from Local Measurements"
_ICML.cc/2025/Conference — ICML 2025 poster_

### Official Review · Reviewer_4FL5 · 2025-03-12

**Overall Recommendation:** 2

**Summary:**

The paper introduces a novel approach termed "explicit circuit representations" for quantum state characterization. Unlike traditional implicit representations, this method allows for direct experimental reconstruction of quantum states. The representations are designed to predict quantum properties accurately based on local measurement data alone. A reinforcement learning-based framework called QCrep is developed to learn these explicit circuit representations. QCrep relies on a local fidelity-based reward function to train an agent, circumventing the barren plateau problem common in gradient-based quantum optimization. The framework uses a Transformer-based measurement feature aggregation block to capture global features of quantum states from local measurements.

**Claims And Evidence:**

1.The claim regarding local and global fidelity is well supported by rigorous proof.
2.The effectiveness of the proposed representation is well supported by numerics.

**Essential References Not Discussed:**

I think the related works are all currently disussed in the paper.

**Experimental Designs Or Analyses:**

I have checked all the experimental designs and they are the main way to validate their proposed framework.

**Methods And Evaluation Criteria:**

Yes.

**Other Comments Or Suggestions:**

1. Obviously, the tables and plots are too small, and hard to read the content inside, which significantly affect the quality of this submission.
2. The structure of the paper is not outlined well, e.g., the Discussion section reads like an extension to the former Experiment section.
3. It is claimed that measuring local observables can mitigate BP; however, this is only supported by a single numerical experiment, with no rigorous proof quantifying the extent of its effectiveness. Moreover, the proposed mitigation approach appears to align with the method described in [1], making the result unsurprising.

[1]. Cerezo, Marco, et al. "Cost function dependent barren plateaus in shallow parametrized quantum circuits." Nature communications 12.1 (2021): 1791.

**Other Strengths And Weaknesses:**

Strengths:
1. The authors introduce local fidelities as reward function, resulting in polynomial sample complexity and claim to mitigate barren plateau.
2. The authors bound the global fidelity well with averaged local fidelity, giving rise to a theoretical guarantee for the RL-based optimization.

Weaknesses/Suggestions are listed below.

**Questions For Authors:**

1. From what it is shown in Figure 1., QCrep framework is a strategy using RL-based method for choosing an appropriate ansatz for quantum state learning (or decoding) at each layer, it looks like an architecture search optimization problem, what’s the core advantage of using transformer-based layers in learning the local measurement data? What’s the sample complexity of the local measurement?
2. When using many-body Hamiltonian for learning task, how is the generalization ability of QCrep in learning the circuit representations in the region of critical behaviors?
3. Several confusions are raised in the comparison of ising model’s evolution. In section 3.2, the time-evolution parameter notation t overlaps the learning step t in the QCrep. And the authors said: “To exhibit the results, we average the performance on different parameters g for each evolution time t.”, is it meaning that QCrep treat the states with different ‘g’ and ‘t’ are notations ‘s’ from the set S, then the framework will learn the circuit representation for each ‘s’, resulting in storing V_{s,t} for them? What is the trotter step when showing first-order Trotter-Suzuki decomposition in Figure 3(b), and what’s the global fidelities of first-order trotter decomposition state?
4. It is with particular interesting to learn a quantum stabilizer codes and their decoding process to respective logical codes. Can QCrep learn the decoding process efficiently? For example, take 5-qubit stabilizer code as training set, how many steps do QCrep need to approximate a perfect decoder?

**Relation To Broader Scientific Literature:**

The application of reinforcement learning to the quantum problem (tomography) falls within the scoop of AI for Science.

**Theoretical Claims:**

Yes, i have checked the correctness of the proof in appendix, i.e. Proposition 2.1.

---

> ### Author Rebuttal · Authors · 2025-03-31
>
> **1. Suggestion 1. Refinement of tables and plots.**
>
> To improve readability, we will refine the tables and figures by moving the legends to the top or embedding them within the figures and abbreviating some metrics in tables. The figures in current paper is in PDF format thus can be scaled larger without loss of resolution.
>
> **2. Suggestion 2. Refinement of the Discussion section.**
>
> The discussion section primarily focuses on evaluating the scalability of our method in more challenging scenarios. The experiments presented in this section shows potential challenges and solutions. We will refine this section by clarifying the topic of discussion and incorporating an exploration of the practical adaptation of our method.
>
> **3. Suggestion 3. Proof on mitigating BP.**
>
> We would like to highlight that our proposed approach is not specifically targeting on mitigating BP, but state characterization (also not tomography). Our main focus is generating circuits for target family of states **locally**, which allows for computing the properties via measuring the output states, and for handling downstream tasks by directly decoding the learned representations without touching the underlying states.
>
> *Therefore, even though the result seems not surprising, it doesn't affect our contribution. In turn, the prior works have provided rigorous guarantee on on mitigating BP with local cost function, which empowers the effectiveness of our reward.* Local cost function has guaranteed trainability for larger systems [1–3]. However, **good trainability does not directly imply good global fidelity and the relation between local cost function and global fidelity remains unclear from the prior works**. Hence, we leverage the advantage of local cost function to design our reward for characterizing large scale quantum systems, and further show how global fidelity can be lower-bounded when the agent learns high local fidelity. Additionally, beyond the numerical experiment we present to demonstrate the mitigation of BP, our extraordinary performance in scaling to much larger systems (e.g., 50 qubits) compared to gradient-based methods also indirectly supports the mitigation of BP.
>
> [1] Sack et al. "Avoiding barren plateaus using classical shadows." PRX Quantum 3.2 (2022): 020365.
>
> [2] Uvarov, A. V., and Jacob D. Biamonte. "On barren plateaus and cost function locality in variational quantum algorithms." Journal of Physics A: Mathematical and Theoretical 54.24 (2021): 245301.
>
> [3] Cerezo, et al. "Cost function dependent barren plateaus in shallow parametrized quantum circuits." Nature communications 12.1 (2021): 1791.
>
> **4. Q1. The advantage of transformer. The sample complexity of local measurement.**
>
> For the advantage of transformer, please refer to the reply 2 to reviewer 9UgW.
>
> A key advantage of our framework over architecture search is that it requires no gradient computation with respect to the circuit parameters, thus can explore more regions during optimization. This feature ensures robustness against noise (see Appendix G), and is natural suitable for tensor network backend, which can simulation large-scale quantum systems. The back-prop from SVD layer is ill-defined when containing zero or repeated singular values if using gradient-based optimizer.
>
> Regarding sample complexity, please refer to Appendix F, and the reply 2 to Reviewer 5m67.
>
> **5. Q2. The generalization ability in the region of critical behaviors for many-body Hamiltonian task.**
>
> To the best of our knowledge, accurately characterizing quantum states in critical region is an open problem of quantum state learning and is not the target of our approach. We use DMRG to simulate many-body ground states, which cannot approximate the real ground state well in the phase transition region. Nonetheless, our method could provide a means to detect phase transition by comparing the fidelity between the reconstructed state and the actual ground state.
>
> **6. Q3. Clarification on the comparison of Ising evolution.**
>
> Your understanding is absolutely correct. There's a misuse of notation of $t$ and we will correct this in a revised version. The trotter step is set to 0.1, as demonstrated in Appendix E. We compare the state fidelity between the learned state and first order trotter decomposed state as shown in Figure 3 (a) and Table 3, where the trotter decomposed state is can approximate the actual evolved state with error rate within 1e-6, but is less efficient if small error is allowed.
>
> **7. Q4. Learning quantum stabilizer codes.**
>
> This is a very interesting topic. However, we note that for a QEC decoder, the output varies according to the input, which is not a quantum state characterization task. It is not compatible with our framework that maps a family of states towards a single state. Extending our framework to support quantum channel characterization could be important yet non-trivial during the short period of rebuttal and we would leave this for future work.

---

### Official Review · Reviewer_9UgW · 2025-03-12

**Overall Recommendation:** 4

**Summary:**

The paper uses a deep reinforcement learning algorithm to construct quantum circuits in a manner that avoids barren plateaus by not requiring gradients with respect to the circuit parameters. The approach appears to outperform alternative approaches, including VQE and QAQA.

A nice addition to the work is the use of a transformer architecture that the authors explicitly tie to the modelling of entanglement between qubits in the circuit. Further, as the circuit is built from local rewards, it appears that the algorithm is scalable.

## Update after rebuttal

I was happy to see the rebuttal from the authors and was happy to raise my score. I am confident that the authors will update the relevant parts of the manuscript and it will make a great contribution to the conference.

**Claims And Evidence:**

The claims the authors wish to explore are well validated. They demonstrate that their algorithm works and, under certain conditions, outperforms alternative approaches.

**Essential References Not Discussed:**

I am not familiar enough with the specific literature to discuss whether essential literature is complete. However, the authors do have a comprehensive state-of-the-art and related work section in both the main manuscript and the supplementary information.

**Experimental Designs Or Analyses:**

I had no issue with the experimental design or analysis.

**Methods And Evaluation Criteria:**

It would be more convincing if the authors discussed/demonstrated scaling to larger circuits or perhaps, more complex problems. However, the chosen benchmarks are relevant.

**Other Comments Or Suggestions:**

Some of the figures were quite small, 6 and 7 in particular.

**Other Strengths And Weaknesses:**

The idea is novel and interesting to the broader community. A possible weakness is the extension to more complex problems but, possibly more importantly, larger circuits. Further, the claims regarding capturing of entanglement were not well validated. Would a simpler architecture have sufficed?

**Questions For Authors:**

1. Can you validate that the transformer architecture is a benefit here? Was there enough data to truly use this complexity and is there any way to explore the capturing of entanglement or correlation effects after training?
2. Can you scale this arbitrarily to large circuit and expect good fidelity?

**Relation To Broader Scientific Literature:**

The approach to building quantum circuits using reinforcement learning is becoming more popular, so this paper aligns well with the emerging field. Further, using transformer architectures to encode entanglement is a novel concept.

**Theoretical Claims:**

There are not many/any theoretical claims made in this manuscript.

---

> ### Author Rebuttal · Authors · 2025-03-31
>
> **1. Suggestion 1. Refinement of figures.**
>
> We will adjust the legend position and font size. Current figures are in PDF format and can be enlarged without loss of resolution.
>
> **2. Q1. Benefits of transformer architecture. Capture of entanglement.**
>
> Compared to simpler architecture like MLP, the transformer in our framework has two main advantages: 1. It allows for capturing entanglement (long-term relation in the measurement data) **without increasing the number of parameters** of the neural network. 2. It enables transfer learning ability to different system sizes. We did further ablation study on learning 50-qubit Ising model to validate our claim. First, we use MLP to replace the transformer block. The input data is flattened and fed into MLP with $49\times 9$ input neurons, which increases with the system size unlike the transformer that is size-agnostic. Note that this formulation doesn't directly allow transfer to a different system size since the architecture is fixed. Second, we perform zero-shot transfer learning of this MLP-based to 10-qubit system, by concatenating 0 to the measurement data. The table below records the performance.
>
> | Model                  | Local               | Global              |
> | ---------------------- | ------------------- | ------------------- |
> | Transformer            | 0.9986 $\pm$ 0.0003 | 0.9673 $\pm$ 0.0083 |
> | Transformer (transfer) | 0.9876 $\pm$ 0.0032 | 0.9391 $\pm$ 0.0160 |
> | MLP-fix                | 0.9985 $\pm$ 0.0002 | 0.9560 $\pm$ 0.0041 |
> | MLP-fix (transfer)     | 0.7311 $\pm$ 0.0271 | 0.1617 $\pm$ 0.0456 |
>
> The local fidelity learned by transformer and MLP are essentially the same, but for global fidelity, transformer outperforms MLP. This reveals that **transformer is better at modeling the correlation in measurement data compared to MLP**. More importantly, transformer enables flexible transferability to quantum system of different sizes.
>
> For architectural complexity, we highlight that our transformer encoder is relatively shallow. It has 2 blocks, each of which contains one 4-head MHA layer and the embedding dimension is only 128. Thus, the architecture is not complex compared with modern LLM. We just utilize its self-attention mechanism, a benefit for modeling quantum data as shown above. More implementation details can be found in the reply 2 to Reviewer cAXj.
>
> **3. Q2. Scalability to larger circuits.**
>
> It's hard to guarantee the scalability to arbitrary large circuit with good global fidelity. Consider a simple case, where your goal is to learn a product state $\bigotimes_{i=1}^n|\psi_i\rangle$. The agent learns to construct a circuit that maps this product state to $|0\rangle^{\otimes n}$ by maximizing the global fidelity, computed as $F=\prod_i |\langle\psi_i|0\rangle|^2$. Suppose the agent is good enough to fit every local term $|\langle\psi_i|0\rangle|^2$ up to accuracy $1-\epsilon$. The total accuracy of $F$ is $(1-\epsilon)^n$, which decays exponentially with the system size. Intuitively, **the difficulty of maintaining good global fidelity would exponentially increase with the system size, even for learning such easy product states**. Nevertheless, in our experiment, we have shown that for learning states like Ising ground states, the performance of zero-shot transfer learning on 70 and 100 qubit systems **remains stable**, as shown in Figure 3(c) and Figure 4(b). This indicates that the agent can capture some underlying pattern of the quantum system that is transferable to another scale, thus we can expect a promising fidelity on relatively large system.
>
> We further did experiments on learning 120 and 150 qubit Ising model. The following table records the results of local and global fidelity.
>
> | System size | Local               | Global              |
> | ----------- | ------------------- | ------------------- |
> | 120         | 0.9985 $\pm$ 0.0006 | 0.9180 $\pm$ 0.0333 |
> | 150         | 0.9984 $\pm$ 0.0007 | 0.8927 $\pm$ 0.0446 |
>
> This shows that even though the fidelity drops with the increase of system size, it doesn't show an exponential degrade, validating the effectiveness of our model.
>
>
> **4. W1. Extension to more complex problems.**
>
> In the following experiment, we consider learning 50-qubit random rotated maximally entangled states (GHZ), where the reduced density matrices are maximally mixed states. The agent has freedom to choose to apply Hadamard, CNOT and $R_z(\theta)$ where $-\pi / 4 < \theta < \pi / 4$. The table below shows the local and global fidelity between the reconstructed and target state.
>
> | Local               | Global              |
> | ------------------- | ------------------- |
> | 0.9927 $\pm$ 0.0151 | 0.9275 $\pm$ 0.1511 |
>
> We can see that **although the states are maximally entangled, the agent can still successfully reconstruct them using the local fidelity reward function if choosing an appropriate action space**. This again highlights the flexibility and applicability of our framework.

---

> > ### Comment · Reviewer_9UgW · 2025-04-02
> >
> > I thank the authors for their comments. I found the results on transferability quite interesting and certainly in favor of the transformers. I would be curious to see how graph networks handled the problem also using attention. In general though, this feedback was helpful. I will raise my score to an accept.

---

### Official Review · Reviewer_cAXj · 2025-03-13

**Overall Recommendation:** 2

**Summary:**

This work develops explicit quantum state representations by generating surrogate preparation circuits through reinforcement learning. The approach uses a local fidelity reward function and a quantum measurement feature aggregation block that extracts global features from local measurement data. The paper attempts to establish theoretical analysis of the relationship between local approximation and global fidelity. Experiments demonstrate successful learning of special classes of quantum states up to 100 qubits.

**Claims And Evidence:**

The claims and evidence are basically fine. One concern is about the implementation of the algorithm, as the loss function involves calculation of the fidelity function of density matrices, which is difficult to estimate. The impact of samples and their inaccurate estimation on the overall method is unclear.

**Essential References Not Discussed:**

The reference is reasonable.

**Experimental Designs Or Analyses:**

The experimental design does not provide sufficient evidence to show whether the method is practical and efficiently implementable for near-term quantum computers.

**Methods And Evaluation Criteria:**

The methods and evaluation criteria are less convincing due to insufficient description of the algorithm details, circuit ansatz, sample cost, and input assumptions.

**Other Comments Or Suggestions:**

The comments are shown above.

**Other Strengths And Weaknesses:**

The work shows an interesting attempt for learning quantum states. But I have some concerns:
1. The algorithm relies on fidelity calculations of density matrices, which are difficult to estimate accurately. As this is a key subroutine, the accuracy vs. sample cost of fidelity needs more analysis. A more careful analysis concerning the whole implementation is needed.
2. When reading the paper, I feel that I need clearer details on the algorithm specifics, circuit ansatz design, optimization, sample complexity, and input requirements or assumptions.
3. For the large-scale solution, the work lacks rigorous proofs to support its claims and theoretical analysis.
4. Experimental results don't sufficiently demonstrate the method's practicality or efficient implementation on near-term quantum hardware.
5. It would be better to provide the detailed methodology of numerical simulations for the experimental part.

**Questions For Authors:**

1. How does your algorithm handle the accuracy-sample cost tradeoff when estimating density matrix fidelity, and what analysis have you done to validate its practical implementation?
2. Could you provide more specific details about your algorithm implementation, circuit ansatz design, optimization procedure, sample complexity requirements, and the input assumptions your method makes?
3. What rigorous proofs support your theoretical claims, particularly for the large-scale quantum state learning solution you propose?
4. What evidence demonstrates that your method is practical and efficiently implementable on near-term or future quantum hardware?
5. Can you elaborate on the detailed methodology used for your numerical simulations (up to which level) in the experimental section?

**Relation To Broader Scientific Literature:**

The relation to broader scientific literature is basically reasonable.

**Theoretical Claims:**

There are no major proofs.

---

> ### Author Rebuttal · Authors · 2025-03-31
>
> **1. W1 & Q1. The accuracy-sample cost tradeoff, estimation of density matrix fidelity.**
>
> The impact of finite sampling (inaccurate expectation value estimation) to the accuracy has been discussed in Appendix G1.
>
> We would like to clarify that while estimating density matrix fidelity is difficult in general, our framework ensures accurate and efficient estimations due to following reasons:
>
> (1) In our framework, instead of evolving the product state 0 towards the target state, we construct circuits to evolve the target state to 0. The two approaches are equivalent because quantum gates are invertible, but our approach only needs to estimate the **local** fidelity between an arbitrary state (density matrix) and 0. This can be done efficiently because it only requires measuring Pauli Z observables.
>
> (2) We only need to compute the local fidelity rather than global fidelity as the reward function for training, which can be estimated by applying local Pauli Z operators to the output state, demonstrating the practical implementation of our framework.
>
> **2. W2 & Q2 & Q5. More implementation details.**
>
> (1) For the **algorithm implementation**, the policy network of the agent contains a 2-layer 4-head Transformer encoder with hidden dimension 128. The positional encoding follows the standard procedure in [1]. The final MLP has 3 linear layers with ReLU activation, and the feature dimension is 512.
>
> (2) The **circuit ansatz design** has been detailed in section 2.3 and section 3, the circuit layout is brickwork shown in Figure 1.
>
> (3) The **input assumptions** to the network has been discussed in section 2.2, where we exclusively measure expectation values of two-local Pauli observables, i.e., XX, XY, YX, YY, YZ, ZX, ZY, ZZ. The impact of inaccurate measurement is discussed in Appendix G1, showing that 1024 measurement shots is enough to obtain a relevantly accurate estimation of expectation values and attain no degrade of performance.
>
> (4) To **optimize** the policy network, we use Adam optimizer with learning rate 0.001. We use stable baselines3 [2] for the implementation of PPO. The batch size is set to 1000. We set a cutoff KL divergence 0.05 between two updates of the network to enhance training stability.
>
> (5) The **numerical simulation details**, including the Tensor Network simulation of quantum systems, and the resource requirement for training and inference are presented in Appendix E and F.
>
> [1] Vaswani, Ashish, et al. "Attention is all you need." Advances in neural information processing systems 30 (2017).
>
> [2] Raffin, Antonin, et al. "Stable-baselines3: Reliable reinforcement learning implementations." Journal of machine learning research 22.268 (2021): 1-8.
>
> **3. W3 & Q3. Proofs for the large-scale quantum state learning.**
>
> Please refer to the reply 3 to Reviewer 9UgW, where we briefly analyze that **even for learning product states, the global fidelity naturally degrades exponentially with the system size, but the performance of our framework remains stable when transferring to larger systems**.
>
> **4. W5 & Q4. Evidence of practicability and efficient implementability on near-term or future quantum hardware.**
>
> First, the measurement settings for our framework is efficient and practical that only require nearest-neighbor measurements, including the input data acquisition and the reward computation. These resources can be easily obtained in real experiments. Second, the action set only contains single-qubit or two-qubit local gates. This means that our framework is implementable even on a quantum hardware without long-term qubit connections. Moreover, our framework follows naturally the spirit of Sim-to-Real [1] learning. The agent could first be trained in the simulation environment, where the expectation values can be accurately and efficiently computed. Then the agent is transferred to a real quantum hardware to conduct state reconstruction.
>
> Furthermore, we have demonstrated the impact of circuit noise in Appendix G2, where the agent is first trained in noiseless environment and transferred to a quantum circuit suffered from depolarizing noise. The output states affected by the
> noise become mixed states. The results shows that the agent can tolerate noise strength below 0.2 and maintain good performance. We anticipate this to be a natural advantage owing to the reinforcement learning pipeline. Because during training, rather than greedily selecting the action that maximizes the reward, the agent makes explorations in different gate combinations and gate parameters, thus becomes aware of how to adjust the action when the observation (the input measurement data) deviates from the ideal case.
>
> [1] Zhao, Wenshuai, Jorge Peña Queralta, and Tomi Westerlund. "Sim-to-real transfer in deep reinforcement learning for robotics: a survey." 2020 IEEE symposium series on computational intelligence (SSCI). IEEE, 2020.

---

### Official Review · Reviewer_5m67 · 2025-03-14

**Overall Recommendation:** 4

**Summary:**

This paper introduces QCrep, a novel reinforcement learning framework for quantum state characterization that generates explicit circuit representations rather than implicit neural encodings. The innovation lies in using local measurements from neighboring qubits to learn circuit descriptions that can faithfully reconstruct quantum states of interest. This represents a significant departure from existing approaches that either require exponentially many measurements or produce black-box neural representations that lack physical interpretability.

The authors developed a transformer-based measurement feature aggregation architecture to extract global quantum features from local measurement data, coupled with a local fidelity reward function that mitigates the notorious barren plateaus problem. Their theoretical analysis establishes a formal relationship between local and global fidelity, providing mathematical justification for their approach. Empirically, they demonstrate QCrep's effectiveness on diverse quantum states up to 100 qubits, including IQP circuit states, time-evolved Ising model states, and many-body ground states, while also showing its utility for downstream tasks like Hamiltonian learning.

**Claims And Evidence:**

The paper's primary claims are generally supported by their theoretical analysis and experimental results:

The claim that QCrep can learn explicit circuit representations for quantum states using only local measurements is convincingly demonstrated across multiple experiments. The authors show they can achieve high fidelity with only $O(N)$ observables instead of the exponential number typically required.

The authors claim their reinforcement learning approach with local fidelity rewards avoids barren plateaus. This is supported indirectly by their ability to scale to systems much larger (50-100 qubits) than what's typically achievable with gradient-based methods, though direct landscape analysis would strengthen this claim.

Their claim regarding zero-shot transfer to different system sizes is compellingly validated by their experiments showing a model trained on 50-qubit systems can generalize to systems ranging from 10 to 100 qubits.

The usefulness of the circuit representations for downstream tasks is demonstrated through Hamiltonian learning experiments, where the learned representations enable parameter prediction with high accuracy.

The paper's evidence for robustness to finite sampling and noise (in the appendix) supports practical applicability, showing the method works with as few as 512 measurement shots and moderate levels of depolarizing noise.

One claim that could use more thorough validation concerns the method's performance on highly entangled states, as the current experiments focus on states with moderate entanglement.

**Essential References Not Discussed:**

The paper's literature review is thorough, but more discussion on relation to NQS could be beneficial

**Experimental Designs Or Analyses:**

The experimental design is comprehensive, covering diverse quantum state families:

For each experiment, the authors clearly describe the system configurations, parameters, and evaluation metrics. The comparisons with baseline methods are fair and thorough.

The zero-shot transfer experiments demonstrating generalization across system sizes are particularly valuable, as is the out-of-distribution generalization test for Heisenberg ground states.

The Hamiltonian learning experiments effectively showcase the downstream utility of the circuit representations, showing they encode physically meaningful information that can be extracted with simple linear models.

The appendix experiments on universal gate sets and mixed-state families demonstrate the method's flexibility beyond the specific configurations in the main paper.

Additional analysis of how performance scales with entanglement complexity would strengthen the experimental component.

**Methods And Evaluation Criteria:**

The paper's methodological approach represents a creative combination of techniques from quantum computing and machine learning. The key insight is inverting the typical state preparation problem: rather than learning circuits that prepare target states from $|0⟩^{\otimes N}$, they learn circuits that evolve target states toward $|0⟩^{\otimes N}$. This enables handling multiple states from a family.

The evaluation criteria are comprehensive and appropriate:

1. Global and local fidelity measure how well the reconstructed states match targets
2. Renyi entropy evaluates the reconstruction of entanglement properties
3. Two-point correlations assess how well quantum correlations are captured
4. Spin-Z values verify the reproduction of local observables

Their comparative analysis against TQS, VQE, QAOA, and QAS consistently demonstrates QCrep's superior performance across all metrics, especially for larger systems.

The ablation studies examining finite sampling effects and circuit noise are crucial for assessing practical viability, though these results would benefit from inclusion in the main paper rather than the appendix.

**Other Comments Or Suggestions:**

No other comments.

**Other Strengths And Weaknesses:**

Weaknesses:

The paper focuses primarily on states with moderate entanglement. The limits of the approach for highly entangled states remain unclear.

The computational complexity analysis could be more thorough, particularly regarding how training costs scale with system size and entanglement.

The paper doesn't fully explore which architectural components contribute most to performance. Additional ablation studies on the model architecture would provide deeper insights.

**Questions For Authors:**

1. How does performance degrade with increasing entanglement entropy of target states? Is there an entanglement threshold beyond which the method becomes ineffective?
2. is it worth exploring more sophisticated reinforcement learning algorithms beyond PPO? Off-policy or model-based RL might further improve sample efficiency.
3. How would using higher n-local fidelity rewards affect results and computational costs? Your theory suggests this could tighten global fidelity bounds.
4. Could the framework be extended to directly optimize for specific quantum properties rather than state fidelity?
5. How does the method perform on mixed states rather than pure states? Many experimental quantum systems produce mixed states due to decoherence.

**Relation To Broader Scientific Literature:**

The paper distinguishes itself from neural quantum states (like GQNQ and NQS) by producing explicit rather than implicit representations, addressing a key limitation of previous ML approaches.

The connection to quantum process tomography could be elaborated further, as circuit representation learning has parallels to process reconstruction.

**Theoretical Claims:**

The paper's theoretical foundation is sound, particularly Proposition 2.1. This provides a mathematical justification for using local fidelity as a reward function.

The proof using spectral decomposition of the local fidelity observable is correct and insightful, showing why local optimization can yield good global properties - a question of fundamental importance in quantum many-body physics.

The theoretical analysis connects to broader questions about the information content of local reduced density matrices and the conditions under which they uniquely determine global states.

The paper doesn't explicitly analyze the representational power of bounded-depth quantum circuits, which would strengthen the theoretical foundation given the empirical success of relatively shallow circuits for complex states.

---

> ### Author Rebuttal · Authors · 2025-03-31
>
> **1. W1. Exploration in highly entangled states.**
>
> Although fully reconstructing the target states for highly entangled states using local fidelity is generally difficult, our framework would work if the state characterization task is to estimate some properties of interests like correlations, which is the primary focus of our proposal. High local fidelity can be obtained even if the target states have higher entanglement like the states generated from random brickwork-like circuits in section 4. Besides, full reconstruction of highly entangled states is not entirely out of reach. For example, prior knowledge on how target states are formed can be leveraged to guide the agent’s learning process. Please refer to reply 4 to Reviewer 9UgW where we learn random rotated GHZ states.
>
> **2. W2. Computational complexity analysis on training cost.**
>
> The training cost depends on the number of observables, the maximum episode length $T$ and the iterations required for convergence. The first two components are analyzed in the main text, where the number of observables scales linearly with the system size regardless of entanglement; 1024 total measurement shots suffice to estimate the expectation values of local observables, and $T$ is fixed for each family of states regardless of system size. The convergence rate, however, is more difficult to analyze theoretically. We provide empirical evidence using the task of learning ground states of the Ising model. We train the agent to learn on 10- and 50-qubit states. The total number of iterations are set to 1740 and 1880. The agent achieves average global fidelity 0.9691 and 0.9673 respectively. This result aligns with the zero-shot transferability of the agent across different system sizes.
>
> **3. W3. Additional ablation studies on the model architecture.**
>
> The policy network is composed of one transformer encoder and one MLP for decision making. For the impact of the transformer encoder, please refer to the reply 2 to Reviewer 9UgW.
>
> **4. Q1. Performance degrade with increasing entanglement entropy.**
>
> For estimating local properties, e.g., Renyi Entropy and two-point correlations, entanglement does not matter much, since the agent can approach high local fidelity for higher entangled states like Figure 7. The degradation of global fidelity is related to the entanglement entropy, if one chooses 1-local fidelity as reward function and chooses a relatively general action space. Consider the Ising evolution as an example, where the circuit depth grows with the evolution time thus entanglement increases. The following table records the scaling of global fidelity with respect to evolution time $t$. The system size is 50 qubits.
>
> | t    | Fidelity |
> | ---- | -------- |
> | 1.2  | 0.9933   |
> | 3.4  | 0.9934   |
> | 5.6  | 0.9816   |
> | 7.8  | 0.6284   |
>
> The performance stays **stable for a relatively long time** before degradation. This is primarily because the bond dimension of the Tensor Network is not enough to accurately represent the state, which is a **limitation of the simulation** rather than our agent. In practice, if we can sample from a large scale quantum computer, we would expect better results.
>
> **5. Q2. Exploring other reinforcement learning algorithms.**
>
> PPO is a **robust and scalable** method that has successfully guided LLM across corpora comprising billions of texts [1], making it a good choice for learning quantum states. While off-policy methods could be explored, the potential inefficiency caused by inferior history actions remains a concern. As for model-based reinforcement learning, our framework already aligns with this paradigm. The simulator we use to evolve the quantum states could be regarded as a surrogate to the real quantum device.
>
> [1] Ouyang, Long, et al. "Training language models to follow instructions with human feedback." Advances in neural information processing systems 35 (2022): 27730-27744.
>
> **6. Q3. Influence of higher n-local fidelity rewards on results and computational costs.**
>
> Higher n-local fidelity will result in barren plateaus problem thus the agent becomes harder or even unable to train if $n$ is too large. The number of observables required scales exponentially with $n$. However, for smaller $n$, e.g., $n\leq 5$, increasing $n$ would help obtain a higher global fidelity in practice, as shown in Figure 7.
>
> **7. Q4. Extendability to directly optimizing specific quantum properties instead of state fidelity.**
>
> Yes. Many properties can be computed by measuring the quantum state using some specific observables like Pauli X, Y and Z. Since the neural network agent can be optimized using local fidelity, obtained by measuring subsystems of the state with Pauli Z, we anticipate that the agent can also be optimized if changing Z to other observables that corresponds to some specific properties of interest.
>
> **8. Q5. Perform on mixed states.**
>
> Please refer to the reply 4 to Reviewer cAXj.

---

> > ### Comment · Reviewer_5m67 · 2025-04-04
> >
> > Thanks for the detailed rebuttal. I would like to keep my recommendation for the paper.

---

### Decision · Program_Chairs · 2025-05-01

**Decision:**

Accept (poster)

**Comment:**

The manuscript received four high-quality reviews and the authors provided rebuttals to all reviewers. Despite an intensive discussion, there is no agreement among the reviewers about the overall assessment: two reviewers argue for a weak reject while two argue for a plain accept. The different perspectives of the reviewers are interpreted by the AC as a sign that the topic of the manuscript is worthwhile being discussed in the whole community. Since none of the reviewers argues strongly to reject the paper and the overall grade is weak accept, the manuscript is accepted.